# ADVERSARIAL SYNTHETIC DATASETS FOR NEURAL PROGRAM SYNTHESIS

## ABSTRACT

Program synthesis is the task of automatically generating a program consistent with a given specification. A natural way to specify programs is to provide examples of desired input-output behavior, and many current program synthesis approaches have achieved impressive results after training on randomly generated input-output examples. However, recent work has discovered that some of these approaches generalize poorly to data distributions different from that of the randomly generated examples. We show that this problem applies to other state-of-the-art approaches as well and that current methods to counteract this problem are insufficient. We then propose a new, adversarial approach to control the bias of synthetic data distributions and show that it outperforms current approaches.

## 1 INTRODUCTION

Program synthesis has long been a key goal of AI research. In particular, researchers have become increasingly interested in the task of programming by example (PBE), where the goal is to generate a program consistent with a given set of input-output (I/O) pairs. Recent studies have achieved impressive results, capable of solving PBE problems that humans would find difficult (e.g., Sharma et al. (2017); Zohar & Wolf (2018); Ellis et al. (2019)). However these studies have a concerning weakness: since large, naturally occurring datasets of program synthesis problems do not exist, these studies train and test their models on synthetic datasets of randomly generated programs and I/O pairs. The justification for using these synthetic datasets is that if a model can correctly predict programs for arbitrary PBE problems, then it has likely learned the semantics of the programming language and can generalize to problems outside the synthetic data distribution (Devlin et al., 2017).

While this justification is plausible, a model might also perform well because it has learned specific aspects of the synthetic data distribution, and recent studies have found this to be the case for several state-of-the-art models (Shin et al., 2019; Clymo et al., 2019). These studies find that current PBE models often perform poorly on distributions different from that of the training data, and they propose methods to mitigate this issue by generating synthetic data with more varied distributions. The idea behind these methods is that a model trained on more varied synthetic data should generalize to a wider variety of distributions, hopefully including those of real-world PBE problems.

Nevertheless, we find that these methods are often insufficient. Previous studies differ on what constitutes a "varied distribution" of synthetic data, creating definitions based on problem-specific heuristics. While generating training data based on these heuristics does help models generalize to certain distributions, we find that models trained using these methods still fail to generalize to many other distributions, including those resembling distributions of real-world problems.

Moreover, different methods fail to generalize to different distributions, raising the question of how one should construct test sets to evaluate these methods. While previous studies have arbitrarily picked test sets that they believe present a reasonable challenge for state-of-the-art methods, this approach may lead to overly optimistic evaluations. A study may report that a method performed well because the researchers failed to find those distributions on which the method performs poorly.

In this paper, we propose an adversarial method to generate a training set. Our adversarial approach builds a training set iteratively, finding data distributions on which a given model performs poorly and adding data drawn from those distributions to the training set on each iteration. We test this method by using it to generate training data for the PCCoder model from Zohar & Wolf (2018), and

we show that models trained using our method generalize to a variety of distributions better than previously proposed methods. Moreover, we propose using a variation of our adversarial approach to generate test sets to evaluate PBE methods. We create test sets for different versions of PCCoder using this approach and show that these test sets reveal weaknesses in models that are not obvious when using other test sets.

This paper makes the following key contributions:

1. We propose a new, adversarial method to generate desirable distributions on which to train models for PBE.

2. We show that models trained using our method generalize to a variety of datasets better than models trained using previously proposed methods.

3. We show that our adversarial approach may also be used to generate test sets that are less likely to overestimate the performance of a model.

## 2 RELATED WORK

Most studies on PBE methods generate I/O pairs using random sampling schemes, filtering out invalid I/O pairs for each program by constraining the sample space and rejecting sets of I/O pairs that do not meet these constraints. Balog et al. (2016) construct a dataset of PBE problems for DeepCoder by enumerating programs up to a given length and removing programs with easily detectable issues (e.g., redundant variables). For each program generated, they then create I/O pairs by sampling inputs uniformly from a restricted range of values guaranteed to yield valid outputs for the given program. Feng et al. (2018) and Zohar & Wolf (2018) also create datasets using the DeepCoder DSL (domain-specific language) in a similar manner.

Bunel et al. (2018) generate PBE problems for Karel (Pattis, 1981) by randomly sampling programs from the Karel DSL and removing programs with obvious problems, similar to Balog et al. They then generate I/O pairs for each program by sampling random inputs and running the program to obtain the corresponding outputs. However, Bunel et al. do not specify what sampling distributions are used for the programs and I/O pairs.

Parisotto et al. Parisotto et al. (2016) create a dataset for the Flashfill domain (Gulwani et al., 2012) by enumerating programs up to 13 expressions long and then randomly sampling inputs to create I/O pairs. They report that while their model achieves 97% accuracy on their synthetic data, they achieve only 38% accuracy on a small dataset of 238 real-world problems. Devlin et al. (2017) use a data generation approach similar to Parisotto et al. but with an improved model and are more successful, achieving 92% accuracy on the same real-world dataset used by Parisotto et al.

All of the papers above focus on advancing models for PBE, but they do so largely using synthetic data to train those models. Shin et al. (2019) report that even minor differences between the synthetic data distributions used for training and evaluation can drastically decrease a model's performance. To solve this problem, they propose a data generation method to improve a model's ability to generalize to other data distributions. They first choose a set of "salient variables" for the domain, defined as a mapping from I/O pairs in the synthetic dataset to a finite, discrete set. They then sample I/O pairs such that the salient variables will be approximately uniformly distributed in the resulting dataset. Shin et al. find that the model proposed by Bunel et al. (2018) better generalizes to a variety of distributions when trained with data generated with this method. However, this method has two major disadvantages. First, it requires the user to determine the correct salient variables, which may be difficult for complex domains. Second, if the domain of valid I/O pairs is highly dependent on the program, it is often prohibitively complex to enforce uniformity across salient variables.

Recently, Clymo et al. (2019) proposed a method to generate PBE problems using a SMT solver. They impose constraints on the I/O pairs to ensure that pairs selected for the dataset are not too similar to each other and then select I/O pairs that satisfy these constraints using a SMT solver. However, when testing an implementation of this method on the DeepCoder domain, the reported improvement of the constraint-based methods over simpler sampling methods is marginal, with the best constraint-based method performing only 2.4% better than the best sampling method. Moreover, many of their constraints are highly specific to the DeepCoder domain, and Clymo et al. do not offer a way to adapt their method to other problem spaces. Nevertheless, they present a method

that does not require the salient variables of Shin et al. (2019), and they show that their approach can be applied to domains such as DeepCoder, for which it is difficult to enforce uniformity across salient variables.

Our adversarial method is closely related to the literature on adversarial training. Most similar to our work is that of Volpi et al. (2018), who propose an adversarial training procedure to learn models resistant to covariate shift. Given a model $M$ trained on data from a given domain $X$, Volpi et al. generate synthetic training data by sampling data points for which $M$ performs poorly, under the constraint that the sampled data points must be within a certain Wassertein distance $\rho$ from $X$. By retraining $M$ on the synthetic data, Volpi et al. create a new model resistant to covariate shifts of magnitude $\rho$. However, a key difficulty in applying their method is that the user must predict the magnitude of a covariate shift before it actually occurs in order to select $\rho$. Moreover, our approach differs from that of Volpi et al. in that our goal is to train a model to generalize to virtually any distribution of data valid for a given programming problem, where as Volpi et al. try to train a model to be resistant to a single, anticipated covariate shift. Our method also does not require the user to predict a hyperparameter such as $\rho$ to train a model.

Other studies have also proposed adversarial methods to generate synthetic training data (e.g., Goodfellow et al. (2015); Sinha et al. (2020)). However, these methods try to train a model to be resistant to imperceptible adversarial attacks, which make small perturbations to training examples such that the perturbed inputs cause the target model to output incorrect answers with high confidence. Our method tries to train a model to generalize to all valid data distributions, not just the small perturbations found in adversarial attacks.

Some studies have proposed iteratively constructing training sets based on the concept of "surprise." In the field of program synthesis, Pu et al. (2017) generate I/O pairs to specify a program by first generating a large set of candidate I/O pairs and then iteratively selecting the I/O pairs from this set that are most surprising, where surprise is estimated using a neural network trained to predict which I/O pairs will be selected (I/O pairs assigned lower probabilities by the neural network are more surprising). Pu et al. use this approach to ensure that their I/O pairs are sufficient to specify desired programs unambiguously. In contrast, our approach tries to find and generalize to out-of-distribution data, and therefore generates I/O pairs to maximize the difficulty of the resulting PBE problems rather than maximizing surprise.

Our adversarial method uses an evolutionary algorithm to find desirable training distributions. This approach is similar to that of studies on competitive co-evolution (e.g., Rosin & Belew (1997); Arcuri & Yao (2007)). For example, Arcuri & Yao (2014) co-evolve populations of programs and populations of unit tests given a formal specification of a desired program. The fitness value of a candidate program depends on how many unit tests it passes, while the fitness value of a candidate unit test depends on how many programs it causes to fail. However, the goal of Arcuri & Yao (2014) and similar works is to simply generate a vanilla program synthesis algorithm capable of generating a program given its formal specification. They do not try to ensure that their algorithms generalize to different data distributions as we do (e.g., their algorithm might only work for certain distributions of formal specifications). Furthermore, in order to compute fitness values, co-evolutionary methods require the user to provide a formal specification of the desired program (e.g., using Z notation (Spivey & Abrial, 1992)). Our approach only requires the user to provide I/O pairs describing the behavior of a desired program, which tend to be easier to provide than a formal specification (Palshikar, 2001).

## 3 METHODOLOGY

We propose an iterative, adversarial approach to build a synthetic dataset of PBE problems. Given a PBE model trained on a given dataset, we use an evolutionary algorithm to find a data distribution on which this model performs poorly. We then add PBE problems drawn from that distribution to the dataset and train a new model on the new, larger dataset. We then repeat the steps above using the new model and new dataset. Appendix A provides pseudocode for this process.

The evolutionary algorithm works as follows. Let $\Phi$ be the space of programs in our DSL, and let $I \times O$ be the space of I/O pairs. Suppose we have a model $M : I \times O \to \Phi$, and we select a family of data distributions of the form $D(\vec{\mu}) : I \to [0, 1]$, where $\vec{\mu}$ are the parameters of the distribution.

We create a population of $n$ distributions $D(\vec{\mu}^i)$, $0 \leq i < n$, with randomly selected parameters $\vec{\mu}^i$. We then draw a set of PBE problems $S_i \subset \{(\phi, i, o) \in \Phi \times I \times O : \phi(i) = o\}$ for each distribution $D(\vec{\mu}^i)$ by sampling $\phi$ uniformly from $\Phi$, sampling $i$ from $I$ according to $D(\vec{\mu}^i)$, and setting $o = \phi(i)$. Next, we calculate fitness scores

$$F(S_i) = \frac{|\{(\phi, i, o) \in S_i : M_{i,o}(i) \neq o\}|}{|S_i|}$$

where $M_{i,o}$ is the program returned by $M$ for $(i, o)$. We choose the $m < n$ distributions $D(\vec{\mu}^i)$ with the highest fitness scores $F(S_i)$. We create a new population of distributions by adding the $m$ fittest distributions to the new population and then for each $D(\vec{\mu}^i)$ within the $m$ fittest distributions, we sample $n/m - 1$ new distributions $D(\vec{\mu}^i + \vec{\epsilon})$ and add those to the new population (where $\vec{\epsilon}$ is a random variable of our choosing). We then repeat the steps above with the new population. Appendix B provides pseudocode for the evolutionary algorithm.

One might be concerned that our evolutionary algorithm could return distributions on which a given model performs poorly due to ambiguity rather than overfitting. For example, a model given I/O pairs of all zeros likely will not return the underlying program, since many programs could potentially create those I/O pairs. However, it is important to note that in the DeepCoder domain, as well as other program synthesis domains, a model given a problem $(\phi, i, o)$, where $\phi(i) = o$, need not return $\phi$. To solve the problem, the model may return any program that maps $i$ to $o$, making ambiguous I/O pairs, such as those containing only zeros, easy to solve. Therefore, we are confident that distributions returned by our algorithm will be those for which the model performs poorly due to overfitting on synthetic data rather than ambiguity of examples.

An obvious disadvantage to our approach is that one must choose the space of distributions to be explored by the evolutionary algorithm. Nevertheless, we believe the task of choosing a space of distributions to be less difficult than the tasks of choosing salient variables (Shin et al., 2019) or SMT constraints (Clymo et al., 2019) necessary for previously proposed methods. Whereas the correct choice of salient variables or SMT constraints may not be intuitive, we argue that the guidelines for choosing a space of distributions for our approach are relatively simple: one should choose the largest space of distributions that can feasibly be searched by the evolutionary algorithm. A larger space of distributions will allow our algorithm to generate more varied training data at the cost of being more difficult to search.

A second disadvantage to our approach is that its runtime is quadratic in the number of iterations, since each iteration requires us to retrain $M$ on a larger and larger training set (assuming the time it takes to train $M$ is linear in the size of the training set). In the experiments below, our approach increased training time from about 12 hours to several days for PCCoder and from 30 minutes to 6 hours for Karel. However, we argue that this increased runtime is not as bad as it may seem. First, there are multiple ways to adjust the runtime. For example, one could generate more PBE problems per iteration and use fewer iterations, or one could decrease the size of $S_i$, decreasing runtime at the cost of increasing the variance of the fitness scores. Furthermore, our approach only increases training time and does not increase the time it takes to synthesize programs for end users.

## 4 EXPERIMENTS

The DeepCoder DSL (Balog et al., 2016) is a language for manipulating lists of integers, capable of expressing complex behaviors such as branching and looping. To evaluate our approach, we generate synthetic data to train PCCoder (Zohar & Wolf, 2018), which solves PBE problems from the DeepCoder domain using a beam search guided by a neural network. In this section, we first discuss the implementation of our adversarial method for the DeepCoder domain and then test this implementation against previously proposed methods.

### 4.1 EVOLUTIONARY ALGORITHM

To implement our evolutionary algorithm, we need to define the space of data distributions our algorithm will explore. The DeepCoder DSL uses integers between -256 and 255 and lists of integers

with lengths between 1 and 20. Therefore, we can define a simple data distribution with four parameters: a lower and upper bound for integers ($a, b : -256 \leq a \leq b \leq 255$) and a lower and upper bound for list lengths ($c, d : 1 \leq c \leq d \leq 20$). For any integer input, we then draw an integer from *discrete uniform*$(a, b)$. For any list input, we draw a list length $l$ from *discrete uniform*$(c, d)$ and then draw $l$ integers from *discrete uniform*$(a, b)$ to fill the list. Our evolutionary algorithm will explore the space of all such distributions.

To calculate the fitness function for a distribution, we draw 100 problems of program length 5 from the distribution and run our model on each problem with a timeout of 60 seconds. When evaluating models in our experiments, we use test sets with program lengths of 5, 8, 10, 12, and 14, and we use a timeout of 300 seconds. However, running models on programs longer than 5 lines with a timeout of 300 seconds would cause our evolutionary algorithm to take prohibitively long to run, so we use shorter programs and timeouts to estimate fitness of distributions.

Next, we define the random variables we will use to mutate our distributions. We mutate integer bounds by adding values drawn from $U(-250/2^i, 250/2^i)$, where $i \in [0, 5]$ is the index of the current iteration of the evolutionary algorithm (starting at 0). We mutate list length bounds by adding values drawn from $U(-20/2^i, 20/2^i)$. Since the bounds of these distributions decrease on each iteration, our evolutionary algorithm is able to explore a wide variety of distributions in earlier iterations and then make smaller adjustments during later iterations.

Finally, we describe how we initialize our population of distributions. As observed by Clymo et al. (2019), smaller input values are valid for more programs in the DeepCoder DSL. Therefore, to speed up training, we initialize $a$ and $b$ from $N(0, 10)$ rather than $U(-256, 255)$ to make it more likely that initial distributions will generate valid I/O pairs. While one may worry that using $N(0, 10)$ will prevent our algorithm from exploring larger values of $a$ and $b$, recall that we initially mutate our distributions with a random variable uniformly distributed between $-250$ and $250$. This allows our algorithm to explore the entire space of available distributions.

One might wonder why our approach adversarially generates inputs but not programs. While one could adapt our evolutionary algorithm to adversarially generate programs as well, we found that manipulating distributions of programs in test sets did not affect the accuracy of PCCoder. Similar to Shin et al. (2019), we created two test sets in which we changed the distributions of programs but left the distribution of I/O pairs identical to that of the original PCCoder training data. In one test set, we did not allow any programs using higher-order functions, and in the other, we did not allow any programs using first-order functions. Neither test set produced any drop in accuracy when used to evaluate PCCoder. Since PCCoder seems to generalize well across different distributions of programs but not different distributions of I/O pairs, we limit our evolutionary algorithm to the generation of inputs. This makes our evolutionary algorithm more efficient, since its search space is smaller.

## 4.2 BASELINES

We compare our approach to three previously suggested approaches. First, we use the approach used by Balog et al. (2016), Feng et al. (2018), and Zohar & Wolf (2018) to generate data for DeepCoder, which we call *Restricted Domain*. This approach imposes a set of program-dependent constraints on inputs that ensure the corresponding outputs are valid. While this approach removes all invalid I/O pairs when generating examples, the constraints are overly strict and remove many valid I/O pairs as well.

Second, we use the approach proposed by Shin et al. (2019), in which we choose a set of what they call "salient variables" and then try to ensure a uniform distribution of those variables in our dataset. We describe Shin et al.'s algorithm to ensure uniformity of salient variables in Appendix C. For our salient variables, we use the same variables that define distributions selected by our evolutionary approach: the minimum and maximum integers of the inputs and the minimum and maximum list lengths of the inputs. We call this the *Salient Variables* approach.

Finally, we use an approach called *Non-Uniform Sampling* proposed by Clymo et al. (2019). For a given program, we generate a value $r$ from an exponential distribution. The input values for I/O pairs are then selected uniformly at random with replacement from the range $[-r, r]$. The intuition behind this approach is that smaller input values are valid for more programs in the DeepCoder DSL,

so sampling with a bias toward smaller values increases the probability that suitable inputs will be found for all programs. Clymo et al. report that they choose a rate parameter of 0.001 for their exponential distribution. However, an exponential distribution with a rate parameter of 0.001 has a mean of 1000, well above the maximum valid value in the DeepCoder DSL of 255. Therefore, we pick of rate parameter of 0.01 for our experiments.

Clymo et al. also propose an approach they call *Semantic Variation*. In this approach they constrain input and output values using constraints that help ensure that I/O pairs selected will be varied. They then use a SMT solver to generate I/O pairs that satisfy these constraints. While we would like to use this approach as a baseline, Clymo et al. do not specify how they encode the constraints that for input $i$ and output $o$ selected for program $\phi$, $\phi(i) = o$ and $i \neq \phi(i)$. This missing detail is important, since Clymo et al. also report that the DeepCoder DSL is too complex to encode such constraints for certain programs $\phi$ and therefore resort to an unspecified approximation of those constraints. However, Clymo et al. report that semantic variation performed only 2.4% better than Non-Uniform Sampling when comparing area under top-$k$ curves, so we believe the inclusion of Non-Uniform Sampling to be sufficient for our experiments.

To compare the above approaches with our own, we generate datasets using each approach and then train the PCCoder model using each dataset, creating four differently trained PCCoder models (three trained using the baselines and one trained using our approach). Each dataset contains 300,000 PBE problems. To create our adversarial dataset, we first train PCCoder on 140,000 problems generated using Restricted Domain as Zohar & Wolf (2018) did for the original version of PCCoder. Using this model for the first iteration, we then add 160,000 problems generated using 20 iterations of our adversarial algorithm, for a total of 300,000 problems (the same number of problems generated for the baselines).

We test our four models on five test sets: three test sets generated using the baselines and two test sets generated from distributions we believe would be common among human-generated examples. We generate the test sets to have the same size and program lengths as Zohar & Wolf (2018): each test set contains 2500 programs of lengths 5, 8, 10, 12, and 14 (with 500 programs of each length).

To create the two test sets generated from "human-like" distributions, we first observe that while no large-scale dataset of human-generated examples exists for the DeepCoder domain, Balog et al. (2016) and Zohar & Wolf (2018) provide several human-created examples of DeepCoder problems in their figures. An immediately noticeable discrepancy between the human-created I/O pairs in the appendices and the synthetic I/O pairs used in those studies is that the human-created I/O pairs use small, positive integers far more frequently than the synthetic data. Moreover, positive integers are more common than negative numbers in the human-created data, regardless of the absolute values of the integers. Based on these observations, we create two "human-like" test sets:

1. **1-to-10**: We generate synthetic data using the constraints from Balog et al. (2016), but we change the valid range for all integers from $[-256, 255]$ to $[1, 10]$.

2. **Small Integers**: We generate synthetic data using the constraints that integer variables must be between 1 and 10 but integers within lists may be between 1 and 255. We remove the constraints of Balog et al. (2016), since these constraints limited the variety of examples in these particular test sets.

When running our models, we set PCCoder's memory size to 11 and its timeout to 300 seconds. We performed all experiments on Intel Xeon 2.30GHz CPUs.

## 4.3 RESULTS

Table 1 shows the percentage of problems solved by each approach. While some of the baselines slightly outperform our adversarial approach on certain test sets–in particular, each baseline unsurprisingly outperforms on its own test sets–the adversarial approach generalizes to all of the baseline test sets reasonably well. Whereas the performance of each baseline approach sometimes drops drastically when evaluated on test sets from other baselines, as shown by the bolded accuracies in Table 1, the adversarial approach performs decently on test sets from all baselines. Moreover, on the human-like test sets, the adversarial approach outperforms all baseline approaches by a significant margin.

Table 1: Performance of PCCoder trained using Restricted Domain, Salient Variables, Non-Uniform Sampling, and our adversarial method. The worst accuracy of each training method for each program length is bolded.

| LENGTH | DATASET | RESTRICTED | SALIENT | NON-UNIFORM | ADVERSARIAL |
|---|---|---|---|---|---|
| 5 | RESTRICTED DOMAIN | 94.6% | 95.4% | 51.8% | 94.0% |
| | NON-UNIFORM SAMPLING | **18.8%** | **21.8%** | 97.4% | 89.0% |
| | SALIENT VARIABLES | 95.8% | 97.6% | 49.8% | 94.0% |
| | 1-TO-10 | 60.4% | 62.0% | **44.0%** | **70.0%** |
| | SMALL INTEGERS | 60.2% | 60.2% | 55.8% | 75.4% |
| 8 | RESTRICTED DOMAIN | 60.4% | 72.6% | **20.8%** | 69.4% |
| | NON-UNIFORM SAMPLING | **17.2%** | **17.8%** | 82.4% | 65.6% |
| | SALIENT VARIABLES | 72.4% | 69.8% | 21.4% | 63.4% |
| | 1-TO-10 | 31.4% | 37.4% | 21.8% | **39.8%** |
| | SMALL INTEGERS | 47.6% | 48.4% | 42.6% | 64.4% |
| 10 | RESTRICTED DOMAIN | 48.8% | 57.8% | 15.2% | 53.0% |
| | NON-UNIFORM SAMPLING | **16.4%** | **16.4%** | 64.8% | 50.0% |
| | SALIENT VARIABLES | 54.2% | 53.2% | **11.6%** | 49.4% |
| | 1-TO-10 | 25.2% | 28.0% | 15.8% | **29.2%** |
| | SMALL INTEGERS | 40.6% | 40.2% | 35.2% | 49.0% |
| 12 | RESTRICTED DOMAIN | 36.8% | 40.2% | **8.0%** | 33.2% |
| | NON-UNIFORM SAMPLING | **6.8%** | **8.6%** | 48.0% | 35.6% |
| | SALIENT VARIABLES | 43.8% | 40.6% | 9.0% | 38.8% |
| | 1-TO-10 | 17.4% | 21.2% | 12.8% | **25.8%** |
| | SMALL INTEGERS | 37.2% | 39.6% | 34.4% | 48.0% |
| 14 | RESTRICTED DOMAIN | 25.0% | 30.2% | **8.6%** | 28.0% |
| | NON-UNIFORM SAMPLING | **8.4%** | **9.4%** | 37.0% | 26.8% |
| | SALIENT VARIABLES | 32.8% | 32.2% | 10.0% | 30.0% |
| | 1-TO-10 | 18.6% | 17.0% | 10.6% | **22.4%** |
| | SMALL INTEGERS | 32.8% | 34.8% | 30.2% | 45.6% |

It is promising to see that the adversarial approach outperforms the baselines on the small integers dataset in particular. The small integers distribution is outside the space of distributions available to our evolutionary algorithm, so in theory, training data generated by our adversarial approach might not improve performance on this distribution. However, our results show that our adversarial approach was able to generate data drawn from distributions similar enough to the small integers distribution to improve performance.

## 4.4 USING THE ALGORITHM TO MEASURE PERFORMANCE

Previous PBE studies have often evaluated the performance of their models using test sets drawn from the same distribution as the training set. However, this approach risks overestimating a model's performance, since a model may perform well on a test set because it has learned features of the synthetic data distribution rather than the semantics of the programming language. Therefore, we argue that a PBE model should be evaluated on a test set drawn from a different distribution than the training set. Naturally, this raises the question of what distribution we should use for the test set. In this section, we propose using our adversarial algorithm to generate test data from those distributions on which the model performs worst. The intuition behind this approach is that adversarially generated test data should come from a distribution as different as possible from the training data by design. If a model can perform well on adversarially generated test data, it has likely learned the semantics of the DSL and is not relying on features specific to the training data.

We use this method to evaluate our four versions of PCCoder. For each version of PCCoder, we use Algorithm 2 to generate a dataset of 100 examples with program length 5. Table 2 shows that our adversarial approach significantly outperforms the 3 baselines. We can also see that there exist distributions on which our models perform far worse than would be suggested by Table 1. For

Table 2: Performance of the four approaches on test sets generated using our evolutionary algorithm. The last column lists the lower and upper bounds for integers and the lower and upper bounds for list lengths for each test set's distribution.

| APPROACHES | % SOLVED | DISTRIBUTION |
|---|---|---|
| RESTRICTED | 6% | -43, 58; 16, 19 |
| NON-UNIFORM | 51% | 2, 55; 18, 20 |
| SALIENT | 9% | -6, 5; 10, 20 |
| ADVERSARIAL | 81% | 2, 12; 4, 17 |

example, while the Salient Variables method achieves an accuracy of at least 20% on all test sets of length 5 in our previous experiments, our evolutionary algorithm finds a distribution for which this method has an accuracy of just 9%.

This evaluation method is far less likely to overestimate performance than those of other studies, which tend to use test sets drawn from the same distribution as the training set or otherwise drawn from arbitrary distributions that the researchers believe will be sufficiently challenging. By testing on arbitrarily chosen distributions, a study may overestimate a model's performance because it overlooked distributions on which the model generalized poorly. However, our approach finds the most difficult distribution for a model within a given search space, providing a more systematic way to test a model's ability to generalize to different distributions.

## 5 EXPERIMENTS ON KAREL

Karel (Pattis, 1981) is an educational programming language that produces instructions for an agent (a robot named Karel) in a rectangular $m \times n$ grid world. Karel creates imperative programs capable of using conditionals and loops, which make the agent move around and alter its grid world. Bunel et al. (2018) propose an algorithm to solve PBE problems for Karel using a beam search guided by a neural network, which Shin et al. (2019) use to test their Salient Variables method. Shin et al. show that changes in the synthetic data distribution severely decrease the accuracy of Bunel et al.'s algorithm; they then use the Salient Variables method to solve this problem, almost completely eliminating the drops in accuracy. In this section, we generate a training set for Karel using our adversarial approach, and we show that our approach eliminates the drops in accuracy just as well as the Salient Variables approach when used to train Bunel et al.'s model. To the best of our knowledge, this makes our approach the first to successfully generalize across multiple program synthesis domains (i.e., DeepCoder and Karel).

### 5.1 EVOLUTIONARY ALGORITHM

Similar to the implementation of our adversarial approach for the DeepCoder domain, we must define the space of data distributions that our evolutionary algorithm will explore. In the Karel DSL, grids can have heights and widths between 2 and 16, and each grid square may contain one of two objects: a marker or a wall (or neither). We define a data distribution with four parameters: a grid height $a$, a grid width $b$, a wall ratio $c$ (the fraction of grid squares containing walls), and a marker ratio $d$ (the fraction of grid squares containing markers). To generate a grid from this distribution, we create an empty $a \times b$ grid. For each grid square, we place a wall in that square with probability $c$. Then, for each grid square without a wall, we place a random number of markers (chosen from *discrete uniform*$(1, 9)$) in that square with probability $d$.

Similar to our implementation for DeepCoder, we draw 100 problems of program length 5 to calculate the fitness function for a distribution. We mutate grid heights and widths by adding values drawn from *discrete uniform*$(-4, 4)$, and we mutate wall and marker ratios by adding values drawn from *discrete uniform*$(-0.25, 0.25)$. Since the space of I/O pairs is far simpler in Karel than in DeepCoder, we find that just two iterations of mutations is sufficient when running our evolutionary algorithm.

## 5.2 RESULTS

We compare our approach to those of Bunel et al. (2018) and Shin et al. (2019). We do not compare our approach to that of Clymo et al. (2019), since there is no clear way to adapt their approach for Karel. We evaluate our three approaches in three ways. First, we use the original test set provided by Bunel et al. Second, we evaluate the approaches on 12 test sets proposed by Shin et al. These test sets have varying distributions of markers and walls, the majority of which are outside the search space of our evolutionary algorithm. We describe Shin et al.'s method to generate these test sets in Appendix D. Finally, we evaluate the three approaches using test sets generated using the adversarial method described in Section 4.4.

On Bunel et al.'s original test set, our method performs marginally better than the Salient Variables method, solving 62.60% of problems compared to Salient Variables' 60.68%. Bunel et al.'s method performs best by a large margin with an accuracy 73.44%. The high accuracy of Bunel et al.'s method is to be expected, since Bunel et al. trained their model on data drawn from the test distribution, allowing their model to overfit on this test set.

On the 12 test sets proposed by Shin et al., both our approach and the Salient Variables approach greatly outperform Bunel et al. The average accuracy of Bunel et al.'s method is 29.61%, while the average accuracy of the Salient Variables method is 62.46%, and the average accuracy of our method is 62.53%. Similarly, when using test sets generated using the method from Section 4.4, Bunel et al. achieve an accuracy of 21.51%, whereas the Salient Variables and adversarial approaches achieve 58.97% and 58.22% respectively. Accuracies of each approach for each individual test set can be found in Appendix E. Clearly, our adversarial method is able to outperform Bunel et al.'s original training method just as well as the Salient Variables method.

However, our method does not outperform the Salient Variables method as it did in the DeepCoder domain. We hypothesize that this is because Karel is a much simpler problem space compared to DeepCoder. In particular, the space of valid I/O pairs is highly program-dependent in DeepCoder but not in Karel. Any valid Karel grid $i$ is a valid input to any Karel program $\phi$, and the output $\phi(i)$ is always guaranteed to be a valid grid as well. In contrast, many potential inputs will be invalid for any given DeepCoder program. For example, if a DeepCoder program takes a list of integers as input, removes all odd numbers from the list, and then returns the first number, then any list containing only odd numbers is an invalid input for that program. Because the space of valid I/O pairs varies from program to program in DeepCoder, it is much harder to sample inputs uniformly with respect to salient variables in DeepCoder than in Karel. Thus the Salient Variables method is able to perform on par with our adversarial method in Karel but struggles in a more complex domain such as DeepCoder. Our adversarial method, however, is able to handle both domains quite well, making it the first such method to generalize across program synthesis domains.

## 6 DISCUSSION

Researchers often train PBE models using randomly generated problems, the idea being that a model that can correctly predict programs for arbitrary inputs has likely learned the semantics of its DSL and can therefore generalize to problems outside the synthetic data distribution. However, recent studies have demonstrated that these models often overfit on their synthetic training sets and often generalize poorly to data from even slightly different distributions. We present a new method to generate synthetic data for program synthesis models that mitigates this issue. We show that a model trained with data generated using this method is able to outperform models trained with previously proposed methods on a variety of baselines.

Moreover, we show that our method can be adapted to generate test sets. Previous studies have chosen distributions for synthetic test sets either by reusing the distribution of the training data or by arbitrarily picking distributions believed to provide a sufficient challenge for their models. We use an evolutionary algorithm to generate test data from distributions that are intentionally as different as possible from the training distribution. These test sets allow for a more conservative evaluation of program synthesis models and allow us to better evaluate a model's ability to generalize to different distributions.

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

## A    ADVERSARIAL ALGORITHM

**Input:** model $M$, dataset $X$, evolutionary algorithm $E$, iterations $N$, episodes $e$, population size $n$, elites size $m$, num probs $c$
**for** $i = 0$ **to** $N - 1$ **do**
   $params = E(M, e, n, m)$
   Randomly generate $c\ programs \subset \Phi$
   Draw $c\ inputs \subset I$ from distribution $D(params)$
   $outputs = [programs[j](inputs[j]), 0 \leq j < c]$
   $problems = zip(programs, inputs, outputs)$
   Add $problems$ to $X$
   Train $M$ on $X$
**end for**

**Algorithm 1:** Adversarial algorithm

## B    EVOLUTIONARY ALGORITHM

**Input:** model $M$, episodes $e$, population size $n$, elites size $m$
Randomly initialize $params = [\vec{\mu}^i, 0 < i < n]$.
**for** $i = 0$ **to** $e - 1$ **do**
   Randomly generate problems $S = [S_j, 0 \leq j < n]$.
   Calculate $fitnessScores = [F(S_j), 0 \leq j < n]$
   Sort $params$ in descending order of $fitnessScores$
   $newParams = params[:m]$
   **for** $j = 0$ **to** $m - 1$ **do**
     **for** $k = 0$ **to** $n/m - 1$ **do**
       Append $newParams[j] + \vec{\epsilon}$ to $newParams$
     **end for**
   **end for**
   $params = newParams$
**end for**

**Algorithm 2:** Evolutionary algorithm

## C    THE SALIENT VARIABLES METHOD

Here we give a more detailed description of the Salient Variables method of Shin et al. (2019) introduced as a baseline in Section 4.2. Let $I$ be the space of inputs, which is originally sampled from a distribution $q : I \to [0, 1]$. Let $X$ be the discrete, finite space of a salient variable, which is calculated by $v : I \to X$. We would like to sample a dataset $D$ from $I$ such that $v$ has a uniform distribution throughout $D$. To do this, we sample $i \sim q$ and calculate $x = v(i)$. We then add $i$ to $D$ with probability $g(i)$, where

$$g(i) = (P_q[v(i) = x] + \epsilon)^{-1} \left( \min_{x' \in X} P_q[v(i) = x'] + \epsilon \right)$$

with $P_q[v(i)]$, the probabilities induced over $v(i)$ via $q$, calculated empirically based on counts computed with past samples drawn from $q$. $\epsilon \in \mathbb{R}^+$ is a hyperparameter that allows the algorithm a tolerance for non-uniformity. Increasing $\epsilon$ increases the algorithm's speed at the cost of allowing the distribution of $v(i)$ to diverge further from the uniform in $D$. We repeat the above until $D$ is of a desired size. Note that $X$ can be a space of tuples, allowing us to ensure that the joint distribution of multiple salient variables is uniform, as we do in Section 4. Algorithm 3 provides pseudocode for this method.

For our implementation in Section 4, we divide our salient variables by 10 and truncate the result. For example, if the minimum integer of an input is -256, we place -26 into our dictionary of counts

in algorithm 3. This way, we have far fewer elements in our dictionary of counts so that $p_{min}$ in algorithm 3 will become greater than 0 after a reasonable amount of time.

---

**Input:** distribution $q$, salient variable $v$, tolerance $\epsilon$, dataset size $n$
Create dictionary of counts $C = \{x : 0, x \in X\}$
D = []
t = 0
**while** $|D| < n$ **do**
    Sample $i \sim q$
    $C[v(i)] = C[v(i)] + 1$
    $t = t + 1$
    $p_{min} = \frac{min_{x \in X} C[x]}{t}$
    $p_{curr} = \frac{C[v(i)]}{t}$
    $g = \frac{p_{min} + \epsilon}{p_{curr} + \epsilon}$
    Sample $h \sim Bernoulli(g)$
    **if** $h$ **then**
        Add $i$ to $D$
    **end if**
**end while**

**Algorithm 3:** Salient Variables algorithm

---

## D    SHIN ET AL.'S TEST SETS

Shin et al. (2019) do not provide the 12 test sets used in their paper, but they describe the process to generate these test sets, and we try to follow their methodology as closely as possible. Each test set contains grids with a wall ratio $r_{wall} \in \{0.05, 0.25, 0.65, 0.85\}$, a marker ratio $r_{marker} \in \{.05, 0.25, 0.65, 0.85\}$, and a marker distribution $D_{marker\ count} \in \{G, U, A\}$, where $G$, $U$, and $A$ stand for *Geom*(0.5), *discrete uniform*(1, 9), and $10 - Geom(0.5)$ respectively. As described by Shin et al., we use the following process to generate grids for a test set with wall ratio $r_{wall}$, marker ratio $r_{marker}$, and marker distribution $D_{marker\ count}$:

1. For each of the 2,500 programs in the original test set from Bunel et al. (2018), we generate $n \sim discrete\ uniform(1, 5)$ input grids. It was unclear from Shin et al. (2019) whether we should generate a random number of input grids or 5 input grids. However, generating a random number of grids yielded results closer to those described by Shin et al.

2. For each of the $n$ grids, we sample a grid width and grid height $x, y \sim discrete\ uniform(10, 16)$ and create an empty $x \times y$ grid.

3. For each grid square, we place a wall in that square with probability $r_{wall}$. If all squares have walls, we resample.

4. For each grid square without a wall, we place markers in that square with probability $r_{marker}$.

5. For each grid square chosen to contain markers, we sample $m \sim D_{marker\ count}$ and place $m$ markers in that square.

6. We place the agent in a random, non-walled grid square in a random orientation.

7. After generating all $n$ input grids for a given program, we check whether the input grids exhibit complete branch coverage (i.e., each branch of the program's control flow is executed by at least one of the inputs). If branch coverage is incomplete, we discard all $n$ grids and repeat the steps above.

It is clear that some of Shin et al.'s test sets do not contain all 2,500 programs from the original test set, since they report accuracies such as 69.37% (if their test sets contained all 2,500 programs, then this would imply that their model solved 1734.25 problems, and it is impossible to solve 0.25 of a problem). Therefore, Shin et al. must discard programs for which they repeatedly find incomplete branch coverage in step 7. Shin et al. do not specify how many times we may find incomplete branch coverage before discarding a program, but we choose to discard a program after hitting step

7 1,000 times. This may explain why Shin et al. report higher accuracies for the Salient Variables method than we do: if Shin et al. chose to use fewer than 1,000 retries, they may have discarded more difficult programs, increasing their reported accuracies.

However, this potential difference in the number of retries does not explain why our accuracies for the baseline method are so much higher than the baseline accuracies reported by Shin et al. Shin et al. report an average accuracy of 10.68% for the baseline method. On our versions of Shin et al.'s test sets, the baseline method achieves an average accuracy of 29.61%. We generated several different versions of Shin et al.'s test sets by altering the process above, changing parameters such as the number of retries, the number of grids generated, and the distribution of grid sizes. The baseline method achieved approximately 30% accuracy for all versions. Therefore, our best guess for the cause of Shin et al.'s lower accuracies is that Shin et al. evaluated the baseline method using a neural network that reached a bad local optimum during training. This would explain why Shin et al. observe such inexplicably low accuracies for the baseline method.

## E  KAREL RESULTS

Table 3: Performance of the three approaches (Bunel et al.'s original approach, salient variables, and our adversarial approach) on the 12 test sets proposed by Shin et al. We describe these test sets in Appendix D.

| $r_{wall}, r_{marker}$ | $D_{markercount}$ | BASELINE | SALIENT | ADVERSARIAL |
|---|---|---|---|---|
| 5 | GEOMETRIC | 42.52% | 58.92% | 59.41% |
| | UNIFORM | 32.63% | 59.32% | 60.12% |
| | 10 - GEOMETRIC | 18.77% | 57.83% | 59.48% |
| 8 | GEOMETRIC | 42.64% | 58.60% | 58.88% |
| | UNIFORM | 33.56% | 57.56% | 57.52% |
| | 10 - GEOMETRIC | 24.77% | 58.20% | 58.56% |
| 10 | GEOMETRIC | 30.41% | 61.80% | 61.22% |
| | UNIFORM | 29.31% | 63.05% | 62.19% |
| | 10 - GEOMETRIC | 24.70% | 62.62% | 62.58% |
| 12 | GEOMETRIC | 25.89% | 71.26% | 71.40% |
| | UNIFORM | 26.38% | 70.13% | 70.00% |
| | 10 - GEOMETRIC | 23.71% | 70.27% | 68.96% |

