# OpenReview forum: "Adversarial Synthetic Datasets for Neural Program Synthesis"
_ICLR.cc/2021/Conference — Reject_

### Official Review · AnonReviewer4 · 2020-10-23
**An incremental, adversarial construction of a PBE dataset with the property that synthesis model M trained on that set generalize better**

**Rating:** 6
**Confidence:** 4

**Review:**

- quality : good
- clarity : good
- originality : okay
- significance : okay

This work mainly deal with the issue of I-O selection during training. Typical program synthesis training collects its dataset as follows:
1) select a random program \phi
2) select a set of random inputs I
3) run \phi(x) to get a set of outputs O
4) add the tuple (\phi, I, O) as training data
5) train a model M using the training data

Given a program \phi, there are infinitely many input-outputs that can be used to specify it. Which one should you pick? This work suggests that instead of picking a random set of I, this set of input should be chosen adversarially, conditioned on the current synthesis model M. Under this scheme, one alternates in generating adversarial training datasets D' and appending it to D, and each time we re-fit a new synthesis model M over this dataset. Experiments show this is good in practice.

I especially liked the passage in this paper that uses the adversarial I-O as test set and measure performance with it. It is a very good lower bound on just how bad your synthesis algorithm is.

Some comments and questions for the authors:

1) why stop at being adversarial w.r.t. input I? why can't we also pick \phi adversarially? Imagine we have a PCFG (or any language model for that matter) capable of generating programs, we can easily evolve a PCFG that targets more "tricky" programs to have them in the grammar. in this sense, this paper title is inaccurate in saying the evolution is evolving how to construct synthetic datasets, as we do not evolve over \phi, but only over I. maybe a more appropriate title would be something along the line of evolving adversarial inputs for synthesis.

2) how expensive is this algorithm? presumably this algorithm needs to cycle between fitting M and generating D. One can imagine this is costly if too many rounds happen? Understandibly, this cost is only paid in training time, but it should not be too high – for instance, re-training M after every single (\phi, I, O) addition. If the time takes to fit M linear in the size of D, Is the time complexity quadratic in the number of iterations? Is 20 iterations (for your experiments) chosen for any particular reason?

3) one line of work that is highly relevant is "selecting representative examples for program synthesis (Pu et.al 2018)", where they start with a big set of input-output examples, and iteratively select from the big set the most "surprising" examples, where surprising is estimated using a neural network trained on the task of I-O infilling.
They used representative examples as a pre-process step to program synthesis, but one can imagine adopting it as a baseline in the following sense:
- first train the neural-network that can measure surprisal using I-O infilling tasks
- sample a random program \phi
- sample a large set of input-outputs I-O_large
- start with I-O_repr = {}
- iteratively add i-o from I-O_large that is surprising – conditioned on I-O_repr – to I-O_repr until I-O_repr is of a desired size
- return (\phi, I-O_repr) as training data point

I-O_repr is representative in a sense that it maximally disambiguate equivalent programs that satisfies the previous set of I-O. It would be interesting to see the relationship between the adversarial examples (optimized for hardness) and the representative examples (optimized for minimizing ambiguity).

4) A question just out of curiosity, how can this approach be scaled to an interactive setting, where the size of I-O can vary? Imagine in an interactive synthesis setting, after giving 3 I-O to the synthesizer, the synthesizer M returns a program that satisfies these 3 I-Os, but isn't what the user want. The user adds a 4th I-O as a result. Would you evolve different sets of adversarial training data one for each I-O size? Is the user being adversarial here though? Or is the user being cooperative, how would you model that?



Overall I enjoyed reading this paper. I find the reasoning of using test sets that are off-distribution compelling, and the usage of an adversarially constructed test-set well justified. This work can be made stronger by considering also evolving a good distribution of \phi (or at least some comments on why this was not attempted), and comparing against the work that selects representative set of I-O.

final recommendation:
after looking at the other reviewer's scores and exchanging some more information with the authors (thanks for the great exchange btw), I would maintain my score of 6. I feel this line of work is definitely useful for program synthesis, especially in the analytical aspects (using adversarially generated examples as benchmarking tools and as representative explanations). the comparisons against other genetic programming approaches, I feel, is not too necessary once the paper is framed as "using adversarial examples to instrument / analyze synthesizer behaviours".

---

> ### Author Response · Authors · 2020-11-23
> **Response to review**
>
> We would like to thank the reviewer for their helpful and insightful comments. Our responses to the specific concerns follow.
>
> (1) This work can be made stronger by considering also evolving a good distribution of $\phi$ (or at least some comments on why this was not attempted).
>
> We have added comments on why this was not attempted in Section 4 (Experiments). In short, we found early on that manipulating distributions of programs in test sets did not affect the accuracy of PCCoder. This is consistent with what Shin et al. found when working with Karel. Although their models initially experienced large drops in accuracy when evaluated on test sets with different distributions of programs, they found that this was because certain kinds of programs were intentionally excluded during data generation, and after removing this intentional exclusion, their models were able to generalize well. Since models for both DeepCoder and Karel seem to generalize well across different distributions of programs but not different distributions of I/O pairs, we limit our evolutionary algorithm to the generation of inputs. This makes our evolutionary algorithm more efficient, since its search space is smaller.
>
> (2) How expensive is this algorithm? Is the time complexity quadratic in the number of iterations?
>
> The algorithm is indeed quadratic in the number of iterations. This increased the time it took to generate data and train models from about 12 hours to several days in the case of PCCoder, and from 30 minutes to 6 hours in the case of Karel. However, as you pointed out, this cost is only paid in training time. We have added an explanation of the increased training time at the end of Section 3 (Methodology).
>
> (3) Is 20 iterations (for your experiments) chosen for any particular reason?
>
> There was no particular reason; we just felt 20 iterations would provide a decent training set within a reasonable amount of time.
>
> (4) This work can be made stronger by comparing against the work that selects representative set of I-O.
>
> We have added such a comparison to Section 2 (Related Work).
>
> (5) A question just out of curiosity, how can this approach be scaled to an interactive setting, where the size of I-O can vary?
>
> There have been a number of methods proposed to resolve ambiguity (i.e., if a user gives 3 examples to the model, the model might detect that there are multiple possible programs that satisfy these examples, and then request that the user give the intended output to a fourth input for which the possible programs give different outputs). These methods show great promise in interactive settings, but they assume that the underlying model will find all possible programs that satisfy a set of examples. This may not be true if the model has overfit on its synthetic training data. We believe that this is where our approach comes in: our approach helps ensure that the model is in fact able to find all possible programs, even if the user provides out-of-distribution examples. Thus we wouldn't try to treat the user as adversarial. Instead, we would use our approach to ensure that our model could handle any out-of-distribution examples that users might provide, and we would use an ambiguity resolution approach to facilitate user interactions.

---

> > ### Comment · AnonReviewer4 · 2020-11-24
> > **thanks for the clarifications, some more feedbacks would be nice.**
> >
> > thanks for the engaging response. I would like to raise some more questions.
> >
> > 1.
> > this is surprising, because it seems to indicate that the synthesis solvers are somewhat agnostic to the underlying distribution of programs. is there any reasonings on why this may be true? It is a known fact that the space of "interesting" programs occupies a narrow slice of the space of "all programs", and since neural-networks are distribution sensitive, it would seem to suggest that neural-guided searches would be very sensitive, instead of agnostic, to the distribution of programs.
> >
> > 2 and 3.
> > fair enough.
> >
> > 4.
> > The operative comparison here is not about "surprising" but more about "representativeness". for instance, if a program can be uniquely determined by a minimum of 10 IO examples, provided that these 10 IO examples are chosen carefully (the minimal representative set), then the goal is to find an approximate representative set, of perhaps 20 examples that can also uniquely determine this program.
> >
> > Thus, I am curious of the following question: is the adversarial IO returned by your algorithm representative in a similar sense? For instance, imagine there are 10 IO examples selected in the incrementally-adversarial fashion per your algorithm. If you sample programs from these 10 IO examples (with neurally-guided search and a sufficiently large budget), how many consistent programs can you find? How does the number of consistent programs change if these 10 IO examples are randomly chosen?
> > If you find that the 10 adversarially chosen examples on average produces less number of consistent programs (under sample from a neurally-guided search), it would suggest that using adversarial methods to incrementally generate IO forms a representative set of IO as well. This would have important applications in \emph{machine teaching}, where a system explains to an end-user a specific program P using a set of representative (or in your case, adversarially generated) examples.
> > I think the experiment should be easy to construct, I'm imagining a plot where on the X axis you have number of IO examples, and on the Y axis you have average number of consistent programs. You can show that the IO generated adversarially would cause a faster decrease in the number of consistent programs. This experiment would bolster your paper because you can now make 3 claims about these adversarially generated IOs:
> > a) they produce programs that are more robust to generalization
> > b) they serve as a good lower-bound of synthesis performance
> > c) they are representative in specifying/communicating a unique program
> >
> > 5.
> > good points about the users cannot be too adversarial w.r.t. out-of-distribution IOs (since you've already done it to yourself in training).  It would be interesting to check how adversarial-ness increase as a function of the number of IO examples. clearly, a set of 1 IO is not as adversarial as a set of 10 IOs. Do you study this relationship? In a similar fashion to 4), can we quantify if there's an inflection point w.r.t. the number of IOs that after which, further adding IO no longer makes the synthesis task more difficult?
> >
> > Let me know what you think about these points. To me this paper is less about helping training a synthesizer better by providing a better dataset (as you can see from other reviewers who complained about the narrow scope of this work also share this sentiment), although it is definitely a nice property. To me, this paper is impactful that by using these adversarially generated examples as instrumentations, we can quantify the nature of the synthesis problem, such as number of example needed to ensure representativeness, and a lower-bound of the difficulty of the underlying synthesis problem.

---

> > > ### Author Response · Authors · 2020-11-24
> > > **Interesting points**
> > >
> > > Thank you very much for the insightful response. We enjoyed reading through it
> > >
> > > (1) The hypothesis we're working with right now is that this has to do with the size/complexity of the program space vs the size/complexity of the input space. Many program synthesis domains, including DeepCoder and Karel, have a relatively small space of possible programs and a much larger space of possible inputs. Consider the DeepCoder DSL for example. There are 38 possible functions in the DSL, so there are at most on the order of 10^22 possible programs of length 14 (the longest program length we consider in experiments), and there are actually significantly fewer than that, since many of those programs will not be valid. In contrast, there are 512 integers in DeepCoder, and programs can take up to 3 lists of 20 integers as input. This gives us about 10^162 possible inputs. So one potential explanation is that the naive random sampling used in most studies is sufficient to represent the program space but not the input space, which is many orders of magnitude larger.
> > >
> > > (4) That makes sense. We'll reframe the comparison in Section 2 (Related Work) accordingly. And agreed, that experiment will be quite interesting to carry out.
> > >
> > > (5) That's a good point. We have not studied that relationship yet, and it's certainly an interesting question.
> > >
> > > Thanks again for your insights. We agree that we should likely do more to emphasize the lower bound on difficulty that our method is able to estimate, and we will certainly look into the experiment you've proposed.

---

> > > > ### Comment · AnonReviewer4 · 2020-11-24
> > > > **thanks for clarifications**
> > > >
> > > > 1) ah I see. this is a good response. I think these kinds of raw stats (size of space of programs) would be good to show more prominently in the experiment section(if not already), that way the readers would know right-away the complexity of the synthesis problem.
> > > >
> > > > 4,5,rest) thanks! hope to see these changes in the future :) and good luck!

---

### Official Review · AnonReviewer3 · 2020-10-26
**Too narrow in scope**

**Rating:** 6
**Confidence:** 3

**Review:**

The paper proposes an approach to develop synthetic datasets aimed at training DeepCoder-alike models. They claim that current approaches to generate synthetic datasets do not help coding algorithms to obtain a good generalization. The reviewed literature and the experiments presented back such claim. The approach consists in an _adversarial_ Evolutionary Computation (EC) methodology.

Pros:
- The paper is, in general, well written. The language is clear, and overall structure of the paper is OK.
- The method is sounding, as it departs from proven concepts such as using evolutionary methods as adversarial sample generators for deep models.

Cons:
-  Sec. 3 could be improved. Authors rely too much on formal notation to express their main contribution, with almost no explanation in simple words, or the intuition behind it.
- Although the paper may be solid enough, from a scientific point of view, I consider the scope of the paper too narrow. Authors are proposing a tool to help another tool without too much use other than competitions on synthetic datasets. As the authors state themselves, there are no enough real-life cases for these types of models to perform, and therefore everything is reduced to benchmark tests on toy, artificially generated, datasets.
- i.e., their contribution is minimal from a wider point of view. This is evidenced in their own list of stated key contributions, where they expand a unique contribution (the proposed approach) into three redundant achievements (experimental evaluation of the approach in two different settings).

Other comments:
A major problem with the entire field of Program synthesis within the field of deep learning, is that sometimes overlaps alarmingly with the field of Genetic Programming (GP), and it becomes difficult to really put in perspective if a contribution is original enough, or just a re-painted version of an older idea. This paper is a clear example of such problem. The method the authors are proposing, bears some resemblance to the (competitive) Co-Evolutionary approach of subset sampling used in GP [1,2,3]. But these methods are never mentioned in the related work area, becoming impossible to perform a side by side comparison of the proposed approach against what already exists in other areas; in other words, the paper fails by being too focused on the area of deep learning, while being oblivious to very similar developments in parallel areas.

1. Doucette, J. A., Mcintyre, A. R., Lichodzijewski, P., & Heywood, M. I. (2012). Symbiotic coevolutionary genetic programming: a benchmarking study under large attribute spaces. Genetic Programming and Evolvable Machines, 13(1), 71-101.
2. Chong, S. Y., Tino, P., & Yao, X. (2008). Measuring generalization performance in coevolutionary learning. IEEE Transactions on Evolutionary Computation, 12(4), 479-505.
3. Pagie, L., & Hogeweg, P. (1997). Evolutionary consequences of coevolving targets. Evolutionary computation, 5(4), 401-418.

Rate updated 30 Nov.

---

> ### Author Response · Authors · 2020-11-23
> **Response to review**
>
> We would like to thank the reviewer for their helpful and insightful comments. Our responses to the specific concerns follow.
>
> (1) I consider the scope of the paper too narrow. Authors are proposing a tool to help another tool without too much use other than competitions on synthetic datasets. As the authors state themselves, there are not enough real-life cases for these types of models to perform.
>
> Many methods trained/evaluated using synthetic data have proven useful in real-world applications. For example, early computer vision methods for autonomous driving were trained and evaluated on synthetic data, and researchers spent a great deal of time developing methods to generate this synthetic data. Synthetic-data-based competitions such as TORCS/SCRC [1,2] have been used in hundreds of works on autonomous driving and control systems. Since collecting real-world datasets of actual road images continues to be logistically difficult, researchers continue to develop improvements to synthetic data generation for autonomous driving even today (e.g., [3]).
>
> Collecting large, real-world datasets of programming problems is similarly difficult, and program synthesis will likely continue to rely on synthetic data as autonomous driving does. Program synthesis models such as RobustFill [4] and TF-Coder [5] have already shown that models trained on synthetic data can prove useful in the real world, and much like autonomous driving, program synthesis will likely become more useful as synthetic data improves. Therefore, we believe that contributions that improve our ability to generate synthetic data for program synthesis are very useful and in fact crucial to the field.
>
> Moreover, the reviewer has mainly focused on our proposition to enhance generation of training sets. Our paper includes a second contribution that is just as, if not more, important: our proposed method is able to generate adversarial test sets to evaluate performance of models. We believe this is an important contribution, since many previous studies have overestimated the performance of their models due to their (often unintentional) cherry-picking of test sets, as shown in the paper. This contribution allows for more objective evaluation of models by providing a lower bound on their performance, regardless of whether those models use synthetic data or real-world data.
>
> (2) They expand a unique contribution (the proposed approach) into three redundant achievements (experimental evaluation of the approach in two different settings).
>
> While we understand that evaluating the proposed approach on two domains could be seen as redundant, we felt it was important to show that our approach worked in multiple very different domains. We believe this is important, since previous methods have not been easily adapted across domains.
>
> (3) The method the authors are proposing bears some resemblance to the (competitive) Co-Evolutionary approach of subset sampling used in GP [1,2,3].
>
> We have added a comparison between our approach and competitive co-evolutionary approaches in Section 2 (Related Work).
>
> References:
> [1] Bernhard Wymann, Eric Espié, Christophe Guionneau, Christos Dimitrakakis, Rémi Coulom, and Andrew Sumner. TORCS, The Open Racing Car Simulator. http://www.torcs.org, 2014.
> [2] Daniele Loiacono, Luigi Cardamone, and Pier Luca Lanzi. Simulated car racing championship: Competition software manual. CoRR, abs/1304.1672, 2013.
> [3] Braden Hurl, Krzysztof Czarnecki, and Steven Lake Waslander. Precise synthetic image and lidar (presil) dataset for autonomous vehicle perception. ArXiv, abs/1905.00160, 2019.
> [4] Jacob Devlin, Jonathan Uesato, Surya Bhupatiraju, Rishabh Singh, Abdel-rahman Mohamed, and Pushmeet Kohli.  Robustfill:  Neural program learning under noisy I/O.CoRR, abs/1703.07469, 2017.
> [5] Kensen Shi, David Bieber, and Rashabh Singh. Tf-coder: Program synthesis for tensor manipulations, 2020.

---

### Official Review · AnonReviewer2 · 2020-10-27
**A valuable contribution to the synthetic data generation**

**Rating:** 6
**Confidence:** 3

**Review:**

## Summary:

Synthesis models trained on synthetic datasets of randomly generated programs and corresponding IO pairs often fail to generalize to real-world PBE problems. The models often learn specific aspects of the synthetic data distribution rather than the semantics of the programming language. This work proposes a more principled adversarial approach to generate synthetic data. The training set is built iteratively by finding data distributions on which a given model performs poorly and adding data drawn from these distributions to the training set. The model and the training dataset are built in tandem. The candidate distributions are generated by mutating the current population of distributions. Models trained using this method are shown to generalize to a variety of distributions.

## Positives:

* The problem is this work is addressing an important one and is well situated.

* Novelty: Although the idea of adversarial data augmentation to improve generalization across domains has already been explored in the literature (eg. Volpi et. al.), I haven’t seen this applied in the PBE setting. The current work is agnostic to the architecture of the synthesis model. I would, however, like the authors to contrast their approach with these approaches.

* Clarity: The paper is well written and well motivated. The algorithm is explained well and seems reproducible.

* Results: The experiments are inline with the claim. The proposed method is evaluated on two domains (DeepCoder and Karel) and compared with two state-of-the-art methods. Although the method is not a winner for all the datasets, it generalizes well and has best worst-accuracy across different datasets for various program lengths.

* The work proposes to use adversarially generated test sets for evaluating performance of models. I believe this is an important contribution and will help in avoiding cherry-picking (often unintentional) of test datasets.

## Negatives

* Defining the space of distributions to be explored may be difficult if the DSL involves APIs with the complex semantics. For example, it will be difficult to use the simplistic method of defining distributions for all the literals for the Sygus String PBE domain. Most of the random IO pairs will not have a solution (or a very long solution). I think it will be difficult to adapt the current method to such a highly constrained domain. I would like the authors to comment on this and qualify the results if they agree.

* To bolster the claim, I would have liked additional results on a complex domain (eg. SYGUS String PBE).

* I believe readers will benefit from some discussion on contrasting the current approach to conventional adversarial data augmentation methods (even though the setting is quite different).

## Overall Remark
I believe the method will be a valuable contribution towards the important problem of generating synthetic datasets to train robust models. Despite my apprehension of the applicability to more complex domains, I believe the method can still be employed in conjunction with the other methods.

## Minor comments
* Although the paper claims adversarial training data generation as the main contribution, the title of the paper focuses on the “evolution” aspect of the method.
* As I understand, in section 3, $(\phi, i, o)$, “i” and “o” are input and output vectors and not a single input output pair. Would like the authors to clarify this.
* “We sample n/m -1 new distributions”. Shouldn’t it be “n/m”

## References
* Volpi, Riccardo, et al. "Generalizing to unseen domains via adversarial data augmentation." Advances in neural information processing systems. 2018.
* Alur, Rajeev, et. al. Syntax-guided synthesis. IEEE, 2013.

---

> ### Author Response · Authors · 2020-11-23
> **Response to review**
>
> We would like to thank the reviewer for their helpful and insightful comments. Our responses to the specific concerns follow.
>
> (1) It will be difficult to use the simplistic method of defining distributions for all the literals for the Sygus String PBE domain. Most of the random IO pairs will not have a solution (or a very long solution). I think it will be difficult to adapt the current method to such a highly constrained domain.
>
> Like SyGuS, DeepCoder is a highly constrained domain in which most random IO pairs will not have a solution (or a very long solution). For example, given two random lists of 20 numbers, it is highly unlikely that there exists a program to map one list to the other using only the simple operations (e.g., add 1 to all elements in a list, divide all elements by 2, etc.) in the DeepCoder DSL. The intuition for why our method still works for highly constrained domains is that we do not simply pick random inputs and outputs and hope that a program exists that can map the inputs to the outputs. Instead, we generate a random program $\phi$ and a random input $i$ and use $o = \phi(i)$ as the output. Thus despite the constraints of DeepCoder, it is actually fairly easy to generate valid IO pairs using our method, and we are confident one could adapt this method for SyGuS.
>
> (2) To bolster the claim, I would have liked additional results on a complex domain (eg. SYGUS String PBE).
>
> While we don't yet have results for another domain, our hope is that our answer to (1) will convince you that DeepCoder is in fact a complex, highly constrained domain.
>
> (3) I believe readers will benefit from some discussion on contrasting the current approach to conventional adversarial data augmentation methods (even though the setting is quite different).
>
> We have added such a discussion to Section 2 (Related Work).
>
> (4) Although the paper claims adversarial training data generation as the main contribution, the title of the paper focuses on the “evolution” aspect of the method.
>
> We have changed the title to "Adversarial Synthetic Datasets for Neural Program Synthesis" to better reflect the main contribution.
>
> (5) As I understand, in section 3, “i” and “o” are input and output vectors and not a single input output pair. Would like the authors to clarify this.
>
> Yes, “i” and “o” are input and output vectors.
>
> (6) “We sample n/m -1 new distributions”. Shouldn’t it be “n/m”?
>
> It is n/m - 1 because the m fittest distributions from one population are carried over into the next population.
>
> Please let us know if this helped clarify the questions and concerns, and let us know if there are any more questions.

---

> > ### Comment · AnonReviewer2 · 2020-11-24
> > **Re: Response to review**
> >
> > Thank you for the clarifications.
> >
> > I believe Sygus String PBE would be more challenging than DeepCoder for data generation. The type signatures for higher order combinators like map and filter can rule of many of the programs that are not type consistent. Whereas in Sygus String domain, most of the random type consistent programs will not generate any output due to complex preconditions.
> >
> > Having said that, I still believe the method can useful in conjunction with the other methods. After careful reassessment, I would like to retain my original rating.
> >
> > Thank you for incorporating the suggestions.

---

### Official Review · AnonReviewer1 · 2020-10-28
**Light and specific**

**Rating:** 3
**Confidence:** 3

**Review:**

The paper is proposing to evolve datasets of (inputs, outputs) in the context of programming by example (PBE), where the goal is to infer a computer program that is consistent with the association of (inputs, outputs). The justification of the approach is to figure out instances that would allow to learn a model with PBE that would generalize better, compared to learning from synthetic datasets that are randomly generated. The approach followed consists to use an evolutionary algorithm to generate this new dataset, using the learn model as a guide, by picking the points where the model at hand performs poorly. Such adversarial approach proceed iteratively, adding extra pairs of (inputs, outputs) to the training set, to get it to perform better in situations where it has trouble.

The proposal is relatively straightforward, using an adversarial optimization loop to enhance the training set for better (at generalization) PBE approach. Such solution is in line with many other adversarial approaches. The proposal is quite specific to the context of PBE, and has been tested in a rather synthetic setting (toy problems / synthetic problems). Performance shows good general improvement of the performance. However, the overhead of generating the adversarial samples is not provided, nor the overhead caused by adding more samples to the training set. If we have trained the model with more random samples (same size for all approaches), would the results be the same -- I mean is there any correction made to ensure that we are always comparing approaches with the same number of samples?

The overall proposition is analogous to different proposals made with coevolution in evolutionary algorithms in the years 2000s. Here again, the training set was "co-evolved" with the program inference task in a competitive (adversarial) way, with a population of samples vs. a population of programs, trying to obtain the samples that would mislead the most the programs, in order to infer more robust solutions. See for instance

https://doi.org/10.1162/evco.1997.5.1.1
https://doi.org/10.1162/106365604773955139

Although we are not talking of the same kind of PBE -- evolutionary algorithms vs. PCCoder -- I think linking the current work with this older one would be required, as the idea is not new (but applied to a different PBE approach).

The quality of writing is not as sharp as we would expect for an ICLR paper. The writing stay somewhat vague at times, we would expect more formal and precise presentation. The algorithms presentation in Appendices A and B are not very clear and not very useful to grasp all the details. The presentation of the methods in Sec. 4.1 and 4.2 are very textual, I am far from being convinced that someone who what to reproduce the approach will be able with the information provided. For instance, how is exactly computed the fitness function is not given explicitly, from what I get I would have a hard time to implement it, or at least valid it in its correctness. It appears strange of proposing an optimization solution as the main contribution without explaining in detail the objective function optimized.

In brief, the proposal does not pass the threshold in terms of significance of the contribution, overall quality of the paper and how exhaustive the experimental evaluation has been made.

---

> ### Author Response · Authors · 2020-11-23
> **Response to review**
>
> We would like to thank the reviewer for their helpful and insightful comments. Our responses to the specific concerns follow.
>
> (1) If we had trained the model with more random samples (same size for all approaches), would the results be the same -- I mean is there any correction made to ensure that we are always comparing approaches with the same number of samples?
>
> All approaches used the same number of samples. Both our models and the baselines were trained on 300000 examples, and thus our models did not benefit from more samples. We changed some wording in Section 4 (Experiments) to try to make this clearer.
>
> (2) The overall proposition is analogous to different proposals made with coevolution in evolutionary algorithms in the years 2000s.
>
> We have added a comparison between our approach and co-evolutionary methods in Section 2 (Related Work). In short, our approach is not exactly analogous, since co-evolutionary methods for PBE from the 2000s typically required the user to specify a desired program using a formal specification language such as Z notation in order to compute fitness values. Our approach has no such requirement.
>
> Moreover, the reviewer has mainly focused on our proposition to enhance generation of training sets. Our paper includes a second contribution that is just as, if not more, important: our proposed method is able to generate adversarial test sets to evaluate performance of models. We believe this is an important contribution, since many previous studies have overestimated the performance of their models due to their (often unintentional) cherry-picking of test sets, as shown in the paper. This contribution allows for more objective evaluation of models by providing a lower bound on their performance.
>
> (3) The fitness function is not given explicitly.
>
> The fitness function is given explicitly in Section 3 (Methodology). It is the expression for $F(S_i)$.
>
> (4) The overhead of generating the adversarial samples is not provided.
>
> We have added a description of the runtime of our method to the end of Section 3 (Methodology).
>
> Please let us know if this helped clarify the questions and concerns, and let us know if there are any more questions.

---

### Author Response · Authors · 2020-11-23
**Summary of revised paper**

Thanks to the reviewers for all the helpful suggestions and comments. We have uploaded a new paper revision with the following key changes:

(1) Section 2 (Related Work) now includes comparisons of our approach to co-evolutionary methods, other methods for adversarial data augmentation, and methods that generate I/O pairs based on "surprise."

(2) We have added a paragraph describing the runtime of our algorithm to the end of Section 3 (Methodology).

(3) We have added a paragraph explaining why we choose to adversarially generate inputs but not programs to the end of Section 4.1 (Evolutionary Algorithm).

(4) We have added additional language to Section 4.2 (Baselines) to make it clear that all models were trained on the same number of PBE problems.

---

### Decision · Program_Chairs · 2021-01-07
**Final Decision**

**Decision:**

Reject

**Comment:**

The paper uses adversarial data to improve generalization in Programming By Example (PBE). The reviews were somewhat mixed with some people finding this useful and interesting while others finding it straightforward and unsurprsing. The reviewers were not convinced of the ultimate usefulness of the approach since it is evaluated on toy or synthetic datasets. The clarity of the presentation could also be improved.